# Tunneling anisotropic magnetoresistance driven by magnetic phase transition

X.Z. Chen[1], J.F. Feng[2], Z.C. Wang [3,4], J. Zhang[5], X.Y. Zhong[3], C. Song [1], L. Jin[4], B. Zhang[6], F. Li[1], M. Jiang[1], Y.Z. Tan[1], X.J. Zhou[1], G.Y. Shi[1], X.F. Zhou[1], X.D. Han[6], S.C. Mao[6], Y.H. Chen[6], X.F. Han[2] & F. Pan[1]

The independent control of two magnetic electrodes and spin-coherent transport in magnetic tunnel junctions are strictly required for tunneling magnetoresistance, while junctions with only one ferromagnetic electrode exhibit tunneling anisotropic magnetoresistance dependent on the anisotropic density of states with no room temperature performance so far. Here, we report an alternative approach to obtaining tunneling anisotropic magnetoresistance in α′-FeRh-based junctions driven by the magnetic phase transition of α′-FeRh and resultantly large variation of the density of states in the vicinity of MgO tunneling barrier, referred to as phase transition tunneling anisotropic magnetoresistance. The junctions with only one α′-FeRh magnetic electrode show a magnetoresistance ratio up to 20% at room temperature. Both the polarity and magnitude of the phase transition tunneling anisotropic magnetoresistance can be modulated by interfacial engineering at the α′-FeRh/MgO interface. Besides the fundamental significance, our finding might add a different dimension to magnetic random access memory and antiferromagnet spintronics.

[1] Key Laboratory of Advanced Materials (MOE), School of Materials Science and Engineering, Tsinghua University, Beijing 100084, China. [2] Beijing National Laboratory for Condensed Matter Physics, Institute of Physics, University of Chinese Academy of Sciences, Chinese Academy of Sciences, Beijing 100190, China. [3] Beijing National Center for Electron Microscopy, School of Materials Science and Engineering, Tsinghua University, Beijing 100084, China. [4] Ernst Ruska-Centre for Microscopy and Spectroscopy with Electrons (ER-C), Forschungszentrum Jülich GmbH, 52425 Jülich, Germany. [5] School of Physics and Wuhan National High Magnetic Field Center, Huazhong University of Science and Technology, 430074 Wuhan, China. [6] Institute of Microstructure and Property of Advanced Materials, Beijing University of Technology, Beijing 100124, China. Correspondence and requests for materials should be addressed to C.S. (email: songcheng@mail.tsinghua.edu.cn) or to F.P. (email: panf@mail.tsinghua.edu.cn)

Tunneling magnetoresistance (TMR), which was observed in early 1990s[1, 2], stands out as a seminal phenomenon in the emerging field of spintronics. The TMR is generated by the parallel and antiparallel states of two ferromagnetic electrodes in magnetic tunnel junctions (MTJs), establishing the foundations of storage functionality for magnetic random access memory[3, 4]. It is generally accepted that a large TMR ratio is fulfilled at the expense of increasing structure complexity to ensure the spin-coherent tunneling through interface[1, 2, 5, 6]. Later, tunneling anisotropic magnetoresistance (TAMR) was observed in the MTJs with only one magnetic electrode, e.g., (Ga,Mn)As/GaAs/Au and [Co/Pt]/AlO$_x$/Pt junctions, due to the interplay between the density of states (DOS) of the magnet and magnetization[7–10]. Besides the ferromagnetic system, the TAMR effect was also observed in antiferromagnetic (AFM) MTJs[11–13]. Unfortunately, the TAMR ratio is generally limited at low temperature (100 K)[7–10, 12] or even enhanced to room temperature in the AFM junctions but the TAMR ratio is persistently below <1%[13, 14], limiting its practical application. Apparently, during the ferromagnetic switching for the tunneling effect, only the magnetization direction is changed, where no modulation of intrinsic magnetism is involved, irrespective of one or two magnetic electrodes. Now the research interest is whether there exists an elegant approach via the manipulation of the intrinsic magnetic ground state to manipulate the spin transport, which would provide an alternative opportunity to obtain tunneling magnetoresistance and make the tunneling behavior more designable.

CsCl-ordered FeRh (α'-FeRh) films, show a first-order phase transition from antiferromagnetic (AFM) to ferromagnetic (FM) order, which can be driven by temperature or magnetic field above room temperature[15–19]. Such an AFM-FM transition means a strong variation of magnetic ground state accompanied by a large DOS variation at the Fermi level[18]. Thus, it would be fundamentally transformative if the magnetic phase transition of α'-FeRh was used to drive the tunneling effect. Basically, the AFM-FM transition itself is associated with an obvious change of resistance[17, 19–21], but the current-in-plane geometry is not capable for implementing high-density storage, thus demanding the experimental exploitation of MTJs structure with current-perpendicular-to-plane geometry as the basis of memories with a cross bar structure. Furthermore, considering the low lattice misfit between MgO and α'-FeRh, a MgO (001) substrate is commonly chosen for the deposition of epitaxial α'-FeRh[17, 19], while epitaxial growth of MgO tunneling barrier is highly expected on the top of α'-FeRh bottom electrode, which would be beneficial for achieving sizeable tunneling effect[5, 6].

Here, we demonstrate an α'-FeRh magnetic phase transition TAMR (PT-TAMR) with the ratio up to 20% at room temperature in MTJs with only one α'-FeRh magnetic electrode, and the polarity and magnitude of PT-TAMR are profoundly dependent on the design of the α'-FeRh/MgO interface.

## Results

**Growth and characterizations of the stacks.** α'-FeRh(30 nm)/MgO(2.7 nm)/γ-FeRh(10 nm) sandwich films were grown on MgO(001) substrates by magnetron sputtering. Figure 1a shows a cross-sectional Z-contrast scanning transmission electron microscopy (STEM) image of the stack films, in which the CsCl ordered-α'-FeRh bottom electrode grown at optimized temperature (grown at 300 °C and annealed at 750 °C) exhibits AFM-FM transition while the non-magnetic γ-FeRh top electrode deposited at room temperature behaves as disordered fcc structure without magnetic phase transition[18–21]. Clearly, the α'-FeRh is the functional layer, whereas the γ-FeRh only serves as the top electrode, similar to Pt or Ta in previous junctions with TAMR[7–14]. The use of γ-FeRh as the top electrode is beneficial for the fully epitaxial growth of the sandwich structure and a sharp top interface. Also visible in Fig. 1a is the epitaxial growth of the stack films with the orientation relationship of α'-FeRh(001)[110] // MgO(001)[010] // γ-FeRh(001)[010]. This is consistent with the small lattice mismatch of 0.3% between α'-FeRh [110]-axis (0.299 nm × $\sqrt{2}$) and MgO [010]-axis (0.421 nm)[17], while somehow larger lattice mismatch of 11% between MgO and γ-FeRh (0.374 nm) results in the existence of dislocations at their interface and the stacking without in-plane rotation. Corresponding schematic of crystalline layout is also displayed in Fig. 1a. Transport measurements of the patterned α'-FeRh/MgO/γ-FeRh junctions were carried out in a four-terminal geometry, as presented in Fig. 1b, ensuring that the resistance measured in such a geometry reflects the transport properties of the MTJs, rather than the magnetic electrodes.

**Tunneling behaviors.** We first show the temperature dependent resistance (R–T) of the bottom electrode at different external magnetic fields ($\mu_0 H$) in Fig. 2a. As expected, the resistance jumps from a high-resistance state (HRS) to a low-resistance state (LRS) with increasing temperature, indicating the magnetic phase transition from AFM to FM of the α'-FeRh bottom electrode above room temperature[18]. The transition temperature ($T_t$) drops with a magnitude of ~ 8 K per Tesla as the magnetic field increases from 1 to 7 T. This behavior is quite characteristic for the magnetic phase transition involving α'-FeRh[15, 17]. The magnetic phase transition of the α'-FeRh electrode would bring about the variation of the tunneling effect. The typical resistance-area (RA) product of the α'-FeRh/MgO/γ-FeRh junctions as a function of temperature (RA–T) is presented in Fig. 2b. The curves were recorded with a bias voltage of 5 mV.

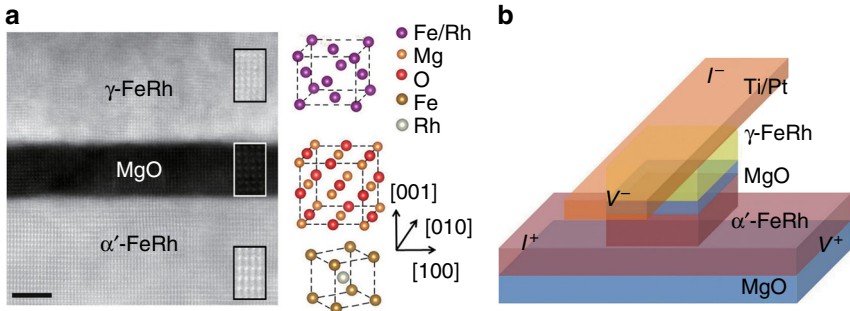

**Fig. 1** Microstructure and measurement geometry of α'-FeRh/MgO/γ-FeRh junctions. **a** Cross-sectional Z-contrast STEM image of the stack films and the schematic of crystal lattice of α'-FeRh, MgO, and γ-FeRh. *Scale bar* is 2 nm in length. **b** A schematic of sample layout and the geometry for four-terminal measurements

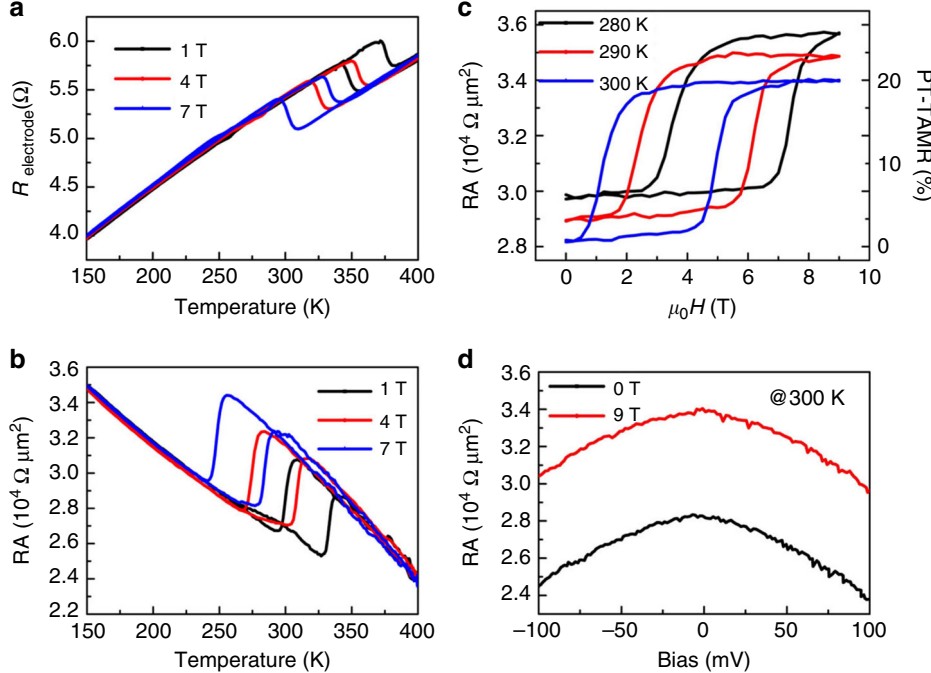

**Fig. 2** Tunneling anisotropic magnetoresistance driven by magnetic phase transition. **a** The temperature dependent resistance (R) of the α′-FeRh bottom electrode at different external magnetic fields. **b** Resistance-area (RA) product of the α′-FeRh/MgO/γ-FeRh junctions as a function of temperature at various external magnetic fields. **c** Field dependent RA curves at several temperatures. The PT-TAMR at 300 K is shown by the right y-axis. **d** Non-collinear behavior of RA versus bias voltage at $H = 0$ for the AFM state and 9 T for the FM state at 300 K

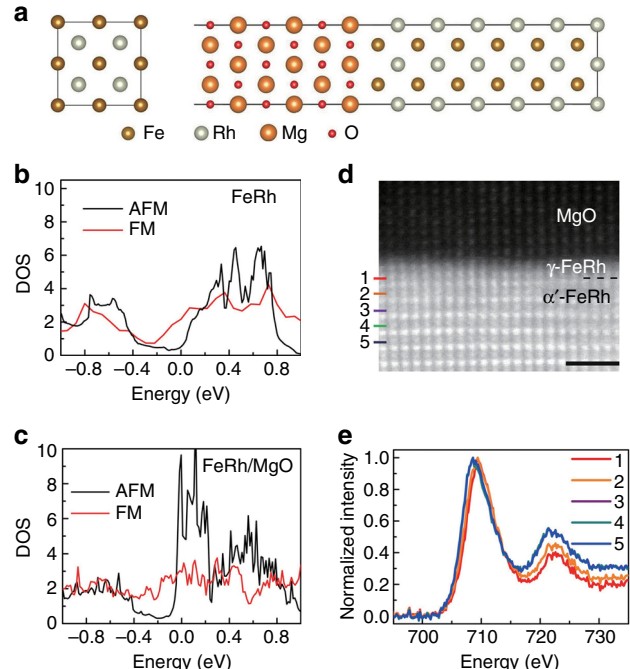

**Fig. 3** Theoretical DOS of α′-FeRh bulk and α′-FeRh/MgO interface, and the interfacial characterization. **a** FeRh and MgO/FeRh supercell with a 20 Å thick vacuum layer were built to calculate the DOS of the bulk FeRh and the interfacial α′-FeRh capped with MgO, respectively. **b, c** Theoretical DOS of α′-FeRh bulk and one unit cell-scaled α′-FeRh in the vicinity of the α′-FeRh/MgO interface. **d** High resolution STEM Z-contrast image with one unit cell-thick γ-FeRh naturally superimposed at the α′-FeRh/MgO interface. *Scale bar* is 1 nm in length. **e** EELS of Fe is marked as 1–5 (marked in **d**) in the order of increasing the distance from the interface with ~0.3 nm gap

The most eminent feature is that a clear first-order phase transition emerges in the RA–T curves, reflecting that the magnetotransport in the MTJs is controlled by the magnetic phase transition of the α′-FeRh bottom electrode. Note that the resistance background of PT-TAMR decreases with increasing temperature, which might be due to the thermal excitations across the barrier[5, 22].

A closer inspection of the RA–T curves shows that a LRS–HRS switching (defined as the positive polarity) is associated with the AFM-FM transition, different from the HRS–LRS switching (the negative polarity) in the α′-FeRh electrode, hence exhibiting an opposite polarity for the junction and the electrode. In addition, a comparison of the transition temperature in Fig. 2b and a reveals that the $T_t$ of the junction is ~ 50 K lower than that of the electrode. This could be explained by the difference between the interfacial and the bulk FeRh: in the MTJ the tunneling behavior is dominated by the interfacial α′-FeRh in the vicinity of the MgO barrier, the $T_t$ of which is reduced (Supplementary Fig. 1), in analogy to the capping effect on the $T_t$ of α′-FeRh films[23].

Magnetic fields provide an alternative approach to triggering the magnetic phase transition. Field dependent RA curves at several temperatures are presented in Fig. 2c. At 300 K, when $\mu_0H$ increases from 4 to 6 T, the tunneling junction undergoes a transition from LRS to HRS, producing a PT-TAMR ratio $PT - TAMR\% = \frac{R_{FM} - R_{AFM}}{R_{AFM}} \times 100\%$ of ~ 20%. This PT-TAMR ratio at room temperature in MTJs with only one α′-FeRh magnetic electrode is more robust than the previously reported AFM-TAMR ratio of ~ 1%[13], let alone no room temperature TAMR in FM systems[7–10]. When the junction is slightly cooled down, the whole resistance background is enhanced, and the transition point increases 1.2 T per 10 K. Nevertheless, the PT-TAMR ratio keeps almost unchanged, revealing that the present PT-TAMR effect is stable, repeatable, and reproducible (Supplementary Fig. 2). We then show in Fig. 2d the bias

**Table 1 Transmission and PT-TAMR ratio of α′-FeRh/MgO/Cu and α′-FeRh/Rh/MgO/Cu junctions**

| Structure | AFM | | FM | | PT-TAMR(%)* |
|---|---|---|---|---|---|
| | $T_{maj}$ | $T_{min}$ | $T_{maj}$ | $T_{min}$ | |
| FeRh/MgO/Cu | $2.8 \times 10^{-4}$ | $2.8 \times 10^{-4}$ | $5.4 \times 10^{-6}$ | $3.9 \times 10^{-5}$ | +1161 |
| FeRh/Rh/MgO/Cu | $7.8 \times 10^{-6}$ | $5.0 \times 10^{-5}$ | $1.1 \times 10^{-6}$ | $2.1 \times 10^{-4}$ | −73 |

*PT-TAMR ratio is calculated by $PT - TAMR\% = \frac{(T_{major}+T_{min})_{AFM} - (T_{major}+T_{min})_{FM}}{(T_{major}+T_{min})_{FM}} \times 100\%$

Abbreviations: AFM, antiferromagnetic; FM, ferromagnetic; PT, phase transition; $T_{maj}$, transmission of majority-spin channel; $T_{min}$, transmission of minority-spin channel; TAMR, tunneling anisotropic magnetoresistance

dependence of the RA at LRS and HRS with $\mu_0 H = 0$ and 9 T, respectively. Both the non-collinear property of the bias dependent RA curves and the decrease of RA with increasing the bias reaffirm the tunneling behavior of the present junction. In addition, the tunneling resistance for the AFM state at the zero-field is lower than that that of the FM state at 9 T in the shown bias scale, supporting the positive polarity of the present PT-TAMR.

**First-principles calculations and interfacial characterization.** The transport experiments have revealed that the tunneling effect of the α′-FeRh/MgO/γ-FeRh junctions is driven by the magnetic phase transition, but shows an opposite polarity with respect to the α′-FeRh electrode. We now try to understand the origin by the first-principles calculations. Accordingly, a FeRh ($2 \times 2 \times 2$) supercell and MgO ($\sqrt{2} \times \sqrt{2} \times 3$)/FeRh ($2 \times 2 \times 6$) supercell with a 20 Å thick vacuum layer were built for calculations (Fig. 3a). The DOS of bulk FeRh and interfacial α′-FeRh capped by MgO are calculated in the absence of spin-orbit coupling (Supplementary Fig. 3). The DOS of the AFM state in Fig. 3b is lower than that of the FM state at the fermi level (Energy = 0) in bulk FeRh, corresponding to the HRS for the AFM state, e.g., the negative polarity. The situation turns out to be dramatically different for the one unit cell (u.c.) of nearest neighbor interfacial α′-FeRh. Figure 3c displays the total DOS of one Fe and one Rh atom in the nearest neighbor of α′-FeRh/MgO interface, reflecting the critical role of the interfacial magnetic layer on the tunneling effect. Remarkably, the DOS of the AFM state overwhelms its FM counterpart at Fermi level, accounting for the lower tunneling resistance in the AFM state compared to the FM case, i.e., the positive polarity (Fig. 2b).

We now focus on the microstructure and chemical information at the α′-FeRh/MgO interface. Figure 3d and e shows an electron energy-loss spectroscopy (EELS) data set acquired using the StripeSTEM technique[24]. Figure 3d shows a high resolution STEM Z-contrast image, from which the EELS spectra were acquired. Surprisingly, one unit cell-thick γ-FeRh is naturally superimposed at the α′-FeRh/MgO interface. This unintendedly ultrathin layer is most likely generated by a part of Fe diffusion into the MgO barrier, which was supposed to occur at their interface[25], and then the Rh-rich composition makes it transform from ordered α′-FeRh to disordered γ-FeRh[26]. Figure 3e shows the fine structure of the Fe $L_{2,3}$ edge, as extracted atomic plane by atomic plane of Fe (marked as 1–5 in Fig. 3d) from the StripeSTEM data set. It is evident that the peak $L_3$ of spectrum 1 shifts to a higher energy direction and the ratio of $L_2/L_3$ increases, suggesting the higher Fe valence at the α′-FeRh/MgO interface. This probably results from the oxidation during the MgO deposition, which is distinct from the stable $Fe^0$ valence at the top MgO/γ-FeRh interface (Supplementary Fig. 4). Spectrum 2 shows the same qualitative behavior, but with a reduced overall magnitude. Differently, spectra 3–5 overlap each other, coinciding with the energy position of $Fe^0$. This characterization reflects that Fe is oxidized to some extent in

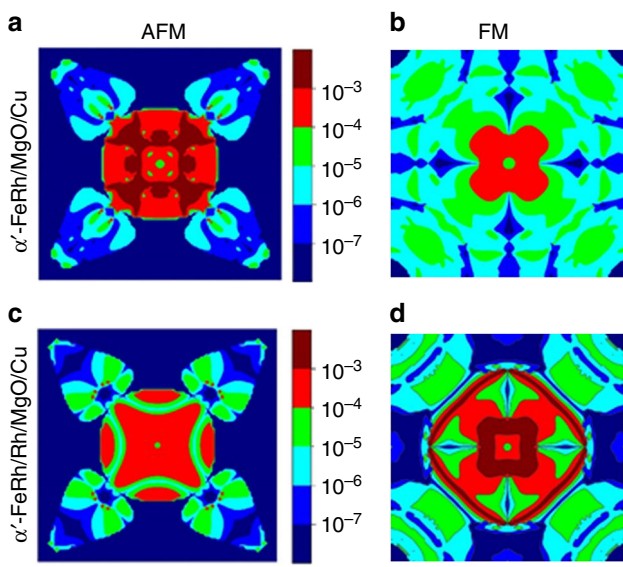

**Fig. 4** Transmission distribution in two-dimensional Brillouin zone for α′-FeRh/MgO(2.5 u.c.)/Cu and α′-FeRh/Rh(1 u.c.)/MgO(2.5 u.c.)/Cu junctions at Fermi level. **a**, **b** Minority-spin channels at AFM and FM states for α′-FeRh/MgO/Cu junctions. **c**, **d** Minority-spin channels at AFM and FM states for α′-FeRh/Rh/MgO/Cu junctions

the superimposed 1 u.c.-thick γ-FeRh. The second unit cell of FeRh from the MgO barrier, i.e., the first α′-FeRh, touches the oxides, which is similar to the case proposed in first-principles calculations. Since the magnetic materials in the vicinity of the tunneling barrier dominates the tunneling effect[1, 2], the magnetic phase transition of oxides neighboring α′-FeRh, whose DOS is inverted by contacting oxides (Fig. 3c) and resultant Fe–O hybridization (Supplementary Fig. 5), plays a profound role on the PT-TAMR. Therefore, α′-FeRh-based MTJs show a positive polarity, in contrast to that of the bulk α′-FeRh, which is also consistent with the lower $T_t$ of the junction compared to its bulk counterpart in Fig. 2.

To quantitatively investigate the PT-TAMR effect induced by magnetic phase transition of α′-FeRh, we performed calculations of transmission distribution in two-dimensional (2D) Brillouin zone for α′-FeRh/MgO(2.5 u.c.)/Cu junctions at Fermi level and the concomitant PT-TAMR ratio. The Cu counter-electrode used here instead of γ-FeRh is to simplify the supercell. The transmission of both minority-spin and majority-spin channels for α′-FeRh/MgO/Cu junctions are listed in Table 1, where the minority-spin channel dominates the transmission at the α′-FeRh/MgO interface, similar to the scenario in Fe/MgO and Fe/GaAs[27, 28]. Accordingly, k-resolved transmission distribution of the minority-spin channel is displayed in Fig. 4. The counterparts for the majority-spin channel are presented in Supplementary Fig. 6. A comparison of the transmission of minority-spin channel at AFM (Fig. 4a) and FM (Fig. 4b) states

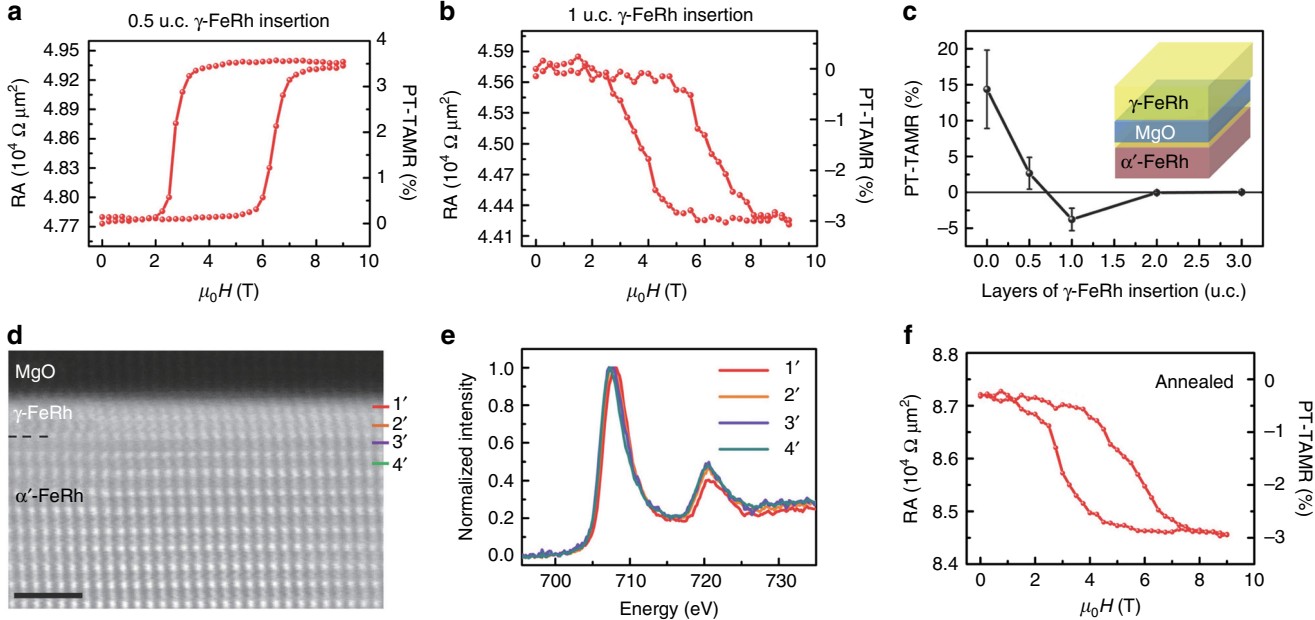

**Fig. 5** Tuning the polarity and magnitude of PT-TAMR via interfacial engineering. **a**, **b** Magnetic field dependent RA of the α′-FeRh-based junctions with intentional insertion of 0.5 u.c. and 1 u.c. γ-FeRh, respectively, between α′-FeRh and MgO during the growth, where the PT-TAMR ratio are 3.5% and −3%, respectively. **c** A summary of the PT-TAMR ratio as a function of the thickness of γ-FeRh insertion, where the schematic of sample layout is also included. The *error bar* is estimated by the s.d. of the measured PT-TAMR in five junctions. The bottom α′-FeRh and tunneling barrier of MgO are denoted in *red* and *blue* layers, respectively. And the insert layer and top electrode of γ-FeRh is shown in *yellow*. **d** High resolution STEM Z-contrast image with 2 u.c.-thick γ-FeRh naturally superimposed at the α′-FeRh/MgO interface in the annealed junctions. *Scale bar* is 1 nm in length. **e** EELS of Fe is marked as 1′–4′ (marked in **d**) in the order of increasing the distance from the interface with ~0.3 nm gap. **f** Typical PT-TAMR curve for the annealed with a PT-TAMR ratio of about −3%

shows that the former is significantly stronger than the later, corresponding to the lower tunneling resistance at AFM state and the positive PT-TAMR. This finding is also consistent to the enhanced DOS at AFM state in Fig. 3c.

The scenario changes dramatically when one unit cell-thick fcc-Rh (1 u.c.) is inserted between α′-FeRh and MgO in α′-FeRh/Rh(1 u.c.)/MgO(2.5 u.c.)/Cu junctions. The intended introduction of 1 u.c. fcc-Rh somehow reflects the main feature of γ-FeRh: Rh-rich composition and no magnetic phase transition. As shown in Fig. 4c, d, the transmission of minority-spin channel at AFM state decreases while the FM case is profoundly enhanced, resulting in the lower transmission at AFM state comparing with its FM counterpart, indicating the reversal of the PT-TAMR polarity with Rh insertion. This change also affirms the critical role of Fe–O hybridization at the α′-FeRh/MgO interface on the observed PT-TAMR effect.

As presented in Table 1, the PT-TAMR ratio for the α′-FeRh/MgO/Cu junction is calculated to be ~1160% with the same sign as the experimental one but with a much higher value, indicating the great potential of the present PT-TAMR by optimizing α′-FeRh/MgO interface. The experimental PT-TAMR ratio in the present case is only ~20%, far below the calculated value, which could be mainly explained by the natural formation of 1 u.c.-thick γ-FeRh at the α′-FeRh/MgO interface. Interestingly, the Rh insertion at the α′-FeRh/MgO interface leads to the reversal of PT-TAMR with the ratio of −73%. The reversal of polarity could be ascribed to the absence of Fe–O hybridization in this scenario (Supplementary Fig. 5).

**Effect of interfacial engineering.** We then turn toward the experimental manipulation of the PT-TAMR effect by interfacial engineering. To check the influence of the emerging γ-FeRh

thickness on the PT-TAMR behavior, 0.5–3 u.c.-thick γ-FeRh grown at room temperature were intentionally inserted between α′-FeRh and MgO. For the insertion of 0.5 u.c.-thick γ-FeRh, the positive polarity of RA–μ₀H curve remains but with a PT-TAMR ratio of ~3.5% (Fig. 5a), much lower than the one without insertion. The scenario differs dramatically when the insertion is up to 1 u.c.-thick γ-FeRh. Concomitant PT-TAMR curve is shown in Fig. 5b, where the PT-TAMR gets reversed from positive to negative. It exhibits the identical polarity as the case with 1 u.c. of Rh insertion in Fig. 4c, d. Accordingly, the PT-TAMR ratio as a function of the thickness of the γ-FeRh insertion is illustrated in Fig. 5c, where the schematic of sample layout is also included. It is found that the PT-TAMR induced by the magnetic phase transition nearly vanishes when the insertion of γ-FeRh is increased to 2 u.c. or above, especially taking the natural existence of 1 u.c.-thick γ-FeRh into account. This tendency could be understood that the interfacial γ-FeRh near the MgO tunneling barrier, which dominates the tunneling effect, has no magnetic phase transition.

As the annealing process could enhance the TMR ratio of traditional Fe/MgO/Fe devices, we now address the question whether sample annealing could also manipulate the tunneling effect of the α′-FeRh/MgO/γ-FeRh junctions. The stack films were annealed in situ at 300 °C for one hour after the deposition of the whole sample. A high resolution STEM Z-contrast image is displayed in Fig. 5d. Remarkably, 2 u.c.-thick γ-FeRh are observed at the α′-FeRh/MgO interface, probably ascribed to more Fe diffusion to the MgO barrier and the resultant Rh-rich phase in a larger scale, compared to its counterpart without annealing. On the other hand, The EELS of Fe at the α′-FeRh/MgO interface in Fig. 5e was recorded using the same method as the one without annealing (Fig. 3d). It is clear that the energy shift of peak $L_3$ in spectrum 1′ remains but the magnitude is

smaller than that without annealing. Meanwhile, the shift in spectrum 2′ is subtle or even negligible, accompanied by no shift in spectra 3′ and 4′. Such a tendency reflects somehow little Fe–O hybridization for α′-FeRh in the annealed sample. Thicker Rh-rich layers and the dense fcc structure of γ-FeRh probably serve as the obstacles for the strong oxygen diffusion. In this case, the α′-FeRh with magnetic phase transition touches the second γ-FeRh unit cell without apparent oxidation, thus producing the same polarity of the PT-TAMR in the annealed MTJs and α′-FeRh bulk. This is bolstered by the PT-TAMR data in Fig. 5f, where a negative PT-TAMR of about −3% is obtained (Supplementary Fig. 7).

Interestingly, both the PT-TAMR ratio and polarity are in a good agreement with the junction with 1 u.c.-thick γ-FeRh insertion (totally 2 u.c.-thick γ-FeRh taking the natural one into account). Their negative polarity could be understood by the blocking of Fe–O hybridization (Supplementary Fig. 5) and resultant lower DOS of the AFM state compared with its FM counterpart, as shown in Fig. 3b. The reduced PT-TAMR ratio could be ascribed to the effective α′-FeRh are 2 u.c. away from the MgO barrier.

## Discussion

The increase of the γ-FeRh insertion results in the reduced PT-TAMR ratio in turn leads us to think about the way to enhance the tunneling effect by removing the γ-FeRh at the α′-FeRh/MgO interface. According to the remarkable transmission difference between the AFM and FM states (Table 1), a larger PT-TAMR ratio of hundreds percent is highly warranted in α′-FeRh-based MTJs if higher quality α′-FeRh/MgO interface was obtained. This might be achieved by optimizing growth parameters or by other growth techniques, e.g., molecular beam epitaxy, to satisfy the requirements of magnetic random access memory on the PT-TAMR ratio. Meanwhile, the memory driven by magnetic phase transition has the potential to be operated in ultrafast dynamics, because the structural evolution of FeRh is faster than the magnetic response[29]. It is also worthy pointing out that temperature variation (Supplementary Fig. 8) or large magnetic field (several Tesla) is not indispensable for the PT-TAMR, because it could be also controlled through electrical means[30], e.g., a fully magnetic phase transition was modulated with a small electric field of 2 kV/cm in FeRh/BaTiO₃ system[21]. The present observation could be also generalized to other materials with magnetic phase transition, such as Fe₃Ga₄[31] and even the transition from G-type to A-type AFM[32].

In summary, the PT-TAMR ratio up to 20% at room temperature in α′-FeRh/MgO/γ-FeRh junctions is driven by the magnetic phase transition of α′-FeRh in the vicinity of the MgO tunneling barrier. The oxygen diffusion into the naturally formed ultrathin (1 u.c.) γ-FeRh at α′-FeRh/MgO interface, making the α′-FeRh contact the oxides, leading to the DOS reversal at the Fermi level for the AFM and FM states. As a result, the junctions show the opposite polarity from the bulk α′-FeRh. Both the γ-FeRh insertion and annealing of stack films, which generate 2 u.c.-thick γ-FeRh at the α′-FeRh/MgO interface, exclude the effect of Fe–O hybridization on the DOS of α′-FeRh, making the junctions show the same polarity of the PT-TAMR as the α′-FeRh bulk, but with reduced magnitude. Thus, our work not only brings about a different approach for the strong PT-TAMR effect but also provides ideas how to manipulate it by designable interfacial engineering.

## Methods

**Sample fabrication**. Bottom magnetic electrodes, 30 nm-thick α′-FeRh, were grown on single crystal MgO (001) by magnetron sputtering at 300 °C and then annealed at 750 °C for 1.5 h. After cooling down to room temperature, a 2.7 nm

MgO barrier was deposited with the base pressure of $4 \times 10^{-7}$ Pa, followed by the capping of 10 nm γ-FeRh at room temperature. Samples with γ-FeRh insertion at the α′-FeRh/MgO interface was grown before the MgO barrier at room temperature with the rate of 0.1 Å/s. In situ annealing was carried out at 300 °C for 1 h. Subsequently, stack films were patterned into rectangle-shaped MTJs pillars of dimensions $5 \times 3$–$30 \times 20$ μm², using optical lithography combined with Ar ion milling and lift-off process. Four-terminal transport measurements are carried out in Physical Property Measurement System.

**Electron microscopy**. High resolution Z-contrast scanning transmission electron microscopy and electron energy-loss spectroscopy were carried out on an FEI Titan 80–300 electron microscope equipped with a monochromator unit, a probe spherical aberration corrector, a post-column energy filter system (Gatan Tridiem 865 ER) and a Gatan 2k slow scan CCD system, operating at 300 kV[33], combining an energy resolution of ∼ 0.6 eV and a dispersion of 0.2 eV per channel with a spatial resolution of ∼ 0.08 nm.

**Calculation**. Density functional theory calculations were performed using the Vienna ab initio Simulation Package. The generalized gradient approximation within the projector augmented-wave method was used for the exchange-correlation interaction. The Brillouin zone is sampled by the k meshes of $24 \times 24 \times 24$ and $24 \times 24 \times 1$ in the Monkhorst-Pack scheme for FeRh ($2 \times 2 \times 2$) and MgO ($\sqrt{2} \times \sqrt{2} \times 3$)/FeRh ($2 \times 2 \times 6$) supercell, respectively. The cutoff energy is 500 eV in wave function expansions. The convergence with respect to cutoff energy and number of k points was carefully checked[34]. Besides, Fe termination at the FeRh/MgO interface is found to occupy a lower energy than the case of Rh termination in the FeRh/MgO supercell. The transmission in two-dimensional Brillouin zone with $100 \times 100$ k points and PT-TAMR are calculated by using Quantum-Espresso package[35] with PBE exchange-correlation potential and ultra-soft pseudo-potential (USPP). In order to calculate the tunneling conductance and avoid using the complicated structure of alloy fcc-FeRh, bcc-Cu electrode is used as the counter-electrode. And also in order to investigate the interlayer effect of FeRh, instead of Rh-rich fcc-FeRh, 1 u.c. fcc-Rh has been inserted between FeRh and MgO interface.

**Data availability**. The data supporting these findings are available from the corresponding author on request.

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

## Acknowledgements

This work made use of the resources of the Peter Grünberg Institute and Ernst Ruska-Centre for Microscopy and Spectroscopy with Electrons in Forschungszentrum Jülich. C.S. acknowledges the support of Young Chang Jiang Scholars Program, Beijing Advanced Innovation Center for Future Chip (ICFC), and Z.C.W. thanks the support of the program of "Strategic Partnership RWTW-Aachen University and Tsinghua University". J.F.F. acknowledges the Youth Innovation Promotion Association of Chinese Academy of Sciences (No. 2017010). This work was financially supported by the National Natural Science Foundation of China (Grant nos. 51671110, 51231004, 51571128, and 51322101) and Ministry of Science and Technology of the People's Republic of China (Grant nos. 2016YFA0203800, 2016YFB0700402, and 2015CB921700).

## Author contributions

C.S. and F.P. conceived and directed the project. X.Z.C., J.F.F., and M.J. prepared the samples. X.Z.C., Y.Z.T., X.J.Z., G.Y.S., X.F.Z. carried out the measurements. Z.C.W., X.Y.Z., L.J., B.Z., X.D.H., S.C.M., Y.H.C. performed the STEM and EELS characterizations, X.Z.C., J.Z. and F.L. carried out the first-principle calculations. X.F.H. provided advice on the experiments. All of the authors participated in discussing the data and writing the manuscript.

## Additional information

**Competing interests:** The authors declare no competing financial interests.

