## [Peer Review File · Nature Communications]

Reviewers' comments:

Reviewer #1 (Remarks to the Author):

The manuscript proposed a novel type of junction with a SINGLE ferromagnetic electrode which exhibits relatively large (~20%) tunneling anisotropic magnetoresistance (TAMR) at room temperature when compared to previous systems typically showing <1% effect below room temperature. The effect relies on the magnetic phase transition in α' -FeRh electrode.

TAMR is of interest to basic physics since it can probe interfacial electronic structure and spin-orbit coupling effects, as well as for applications in spintronics since it offers detection of magnetic fields without using traditional two ferromagnetic electrodes where fixing the magnetization for one of them can be quite cumbersome in practice. This makes manuscript publishable.

However, the authors should clarify the following:

1. The manuscript is focused on TAMR, but title, abstract and various labels use TMR (=tunneling magnetoresistance) which is VERY confusing since it suggests conventional magnetoresistance in junction with two ferromagnetic layers. In fact, the authors do use γ -FeRh as the second layer, so only after careful reading one realizes that that layer is actually paramagnetic. Therefore, the title and notation should be changed to TAMR.

2. There is very little theoretical calculations supporting the results, and without any technical details about what is actually calculated. For example, it is not clear what kind of density of states (DOS) is computed in Fig. 3c - is it local DOS at some plane within interface region, or local DOS integrated over some region of space around the interface. Typically, spin-orbit coupling plays a crucial role for TAMR, which could have been checked for the device in the present manuscript by performing calculations in Fig. 3 with spin-orbit turned ON and OFF. The proper handling of these two issues is illustrated by calculations in, e.g., Physical Review Letters 98, 046601 (2007).

Reviewer #2 (Remarks to the Author):

Manuscript NCOMMS-17-01166-T entitled "Tunneling magnetoresistance driven by magnetic phase transition" by X. Z. Chen et al. systematically analyzes the PT-TMR effect amplitude as a function of the interfacial microstructure of α' -FeRh/MgO combining fascinating high resolution analytical HRTEM with ab initio DOS calculations. Experimentally this paper is very sound and thorough microstructural, magnetic as well as transport investigations consistently support the main finding. The observed PT-TMR effect amplitude of 20% at room temperature is driven by the magnetic phase transition of α' -FeRh in addition to the presence of a naturally formed one-unit cell thick γ -FeRh layer at the α' -FeRh/MgO interface, which results in an opposite polarity as expected for α' -FeRh. Furthermore, the authors present two approaches – annealing and inserting an additional γ -FeRh layer at the α' -FeRh/MgO interface – successfully avoiding the oxidation problematic. Both approaches repair the reversed polarity of the PT-TMR effect amplitude but result in a PT-TMR effect amplitude reduced to about -5% at maximum only. All discussions presented in this manuscript are taking into account all aspects of current and previous literature.

In summary, this is a very decent work but dealing with the common problematic known for all TMR-cells, the affinity for oxygen of elements of the magnetic electrodes sandwiching an oxidic barrier. Since both approaches to avoid the oxygen diffusion are drastically reducing the initial TMR-effect amplitude, I cannot see at all evidence that this finding is adding a different dimension to MRAM and

spintronics.

Hence, my recommendation is not to publish this paper in Nature Communications.

Reviewer #3 (Remarks to the Author):

The authors report interesting experimental results of demonstration of tunneling magnetoresistance (TMR) using the first order magnetic phase transition between antiferromagnetism and ferromagnetism in α' -FeRh at room temperature. They have proved that the electronic structure near the tunneling barrier has a crucial role for determining the TMR ratio. If we can properly control the phase transition of this material near the tunneling interface, this method will be very useful. In my opinion, their idea is promising for the room temperature applications. However, the following issues should be properly addressed in the manuscript for publication in Nature Communications.

1. They did not mention whether this device can have a non-volatile memory function. In Fig. 2c, the resistance area (RA) always returns to the low state when the magnetic field is switched off. For the memory applications, we have to maintain each state without a magnetic field.
2. They did not mention the effect of the strain induced by the phase transition. It may degrade the endurance of the device.
3. I hope that the authors describe the switching speed. I wonder if it is slower than the magnetization switching.
4. The physical mechanism of the change of the density of states (or the electronic structure) by the magnetic phase transition should be explained in more detail. This is important to increase the TMR ratio.
5. L153: Why is the tunnel resistance background changed when the temperature is changed? If the tunneling effect is dominant, it should be constant.

Response Letter of NCOMMS-17-01166A

Response to Reviewer #1

R: The manuscript proposed a novel type of junction with a SINGLE ferromagnetic electrode which exhibits relatively large (~20%) tunneling anisotropic magnetoresistance (TAMR) at room temperature when compared to previous systems typically showing <1% effect below room temperature. The effect relies on the magnetic phase transition in α' -FeRh electrode.

TAMR is of interest to basic physics since it can probe interfacial electronic structure and spin-orbit coupling effects, as well as for applications in spintronics since it offers detection of magnetic fields without using traditional two ferromagnetic electrodes where fixing the magnetization for one of them can be quite cumbersome in practice. This makes manuscript publishable.

A: We appreciate the positive evaluation of Reviewer #1.

R: 1. The manuscript is focused on TAMR, but title, abstract and various labels use TMR (=tunneling magnetoresistance) which is VERY confusing since it suggests conventional magnetoresistance in junction with two ferromagnetic layers. In fact, the authors do use gamma-FeRh as the second layer, so only after careful reading one realizes that that layer is actually paramagnetic. Therefore, the title and notation should be changed to TAMR.

A: As advised by the referee, "TMR (tunneling magnetoresistance)" has been changed to "TAMR (tunneling anisotropic magnetoresistance)" in the title, main text, and supplementary material where necessary.

R: 2. There is very little theoretical calculations supporting the results, and without any technical details about what is actually calculated. For example, it is not clear what kind of density of states (DOS) is computed in Fig. 3c - is it local DOS at some plane within interface region, or local DOS integrated over some region of space around the interface. Typically, spin-orbit coupling plays a crucial role for TAMR, which could have been checked for the device in the presence manuscript by performing calculations in Fig. 3 with spin-orbit turned ON and OFF. The proper handling of these two issues is illustrated by calculations in, e.g., Physical Review Letters 98, 046601 (2007).

A: Thanks for the referee's suggestion and the useful reference. We show in Fig. 3c the total DOS of one Fe atom and Rh atom which are the first neighbor of the α' -FeRh/MgO interface. We add a sentence in Page 9 Line 13: **Figure**

3c displays the total DOS of one Fe and one Rh atom in the nearest neighbor of α' -FeRh/MgO interface.

Furthermore, the DOS with spin-orbit coupling (SOC) On and OFF are calculated and presented in Fig. S3. It is found that the DOS variation with or without SOC is negligible in contrast to the apparent effect induced by magnetic phase transition. This observation reveals that the present PT-TAMR originates from magnetic phase transition and the resultant electronic structure evolution instead of SOC. To quantitatively understand the present PT-TAMR effect, the calculation of transmission distribution and PT-TAMR ratio are carried out with the help of the mentioned reference (Phys. Rev. Lett. 98, 046601 (2007)). Interestingly, the positive and negative PT-TAMR ratios are obtained for α' -FeRh/MgO/counter-electrode and α' -FeRh/Rh/MgO/counter-electrode junctions (Table 1), respectively, which coincides with the experimental findings (Figs. 2 and 5).

Accordingly, we add Fig. 4, Table 1 to the main text and Figs. S3 and S6 to Supplementary material. Corresponding discussions could be found in Page 11–13.

Figure 4 | Transmission distribution in two dimensional Brillouin zone for α' -FeRh/MgO(2.5 u.c.)/Cu and α' -FeRh/Rh(1 u.c.)/MgO(2.5 u.c.)/Cu junctions at Fermi level. a,b, minority-spin channels at AFM and FM states for α' -FeRh/MgO/Cu junctions. c,d, minority-spin channels at AFM and FM states for α' -FeRh/Rh/MgO/Cu junctions.

To quantitatively investigate the PT-TAMR effect induced by magnetic phase transition of α' -FeRh, we performed calculations of transmission distribution in two dimensional Brillouin zone for α' -FeRh/MgO(2.5 u.c.)/Cu junctions at Fermi level and the concomitant PT-TAMR ratio. The

Cu counter-electrode used here instead of γ -FeRh is to simplify the supercell. The transmission of both minority-spin and majority-spin channels for α' -FeRh/MgO/Cu junctions are listed in Table 1, where the minority-spin channel dominates the transmission at the α' -FeRh/MgO interface, similar to the scenario in Fe/MgO and Fe/GaAs^{27,28}. Accordingly, *k*-resolved transmission distribution of the minority-spin channel is displayed in Fig. 4. The counterparts for majority are presented in Supplementary Fig. S6. A comparison of the transmission of minority-spin channel at AFM (Fig. 4a) and FM (Fig. 4b) states shows that the former is significantly stronger than the later, corresponding to the lower tunneling resistance at AFM state and the positive PT-TAMR. This finding is also consistent to the enhanced DOS at AFM state in Fig. 3c.

The scenario changes dramatically when one unit cell-thick fcc-Rh (1 u.c.) is inserted between α' -FeRh and MgO in α' -FeRh/Rh(1 u.c.)/MgO(2.5 u.c.)/Cu junctions. The intended introduction of 1 u.c fcc-Rh somehow reflects the main feature of γ -FeRh: Rh-rich composition and no magnetic phase transition. As shown in Fig. 4c and d, the transmission of minority-spin channel at AFM state decreases while the FM case is profoundly enhanced, resulting in the lower transmission at AFM state comparing with its FM counterpart, indicating the reversal of the PT-TAMR polarity with Rh insertion. This change also affirms the critical role of Fe-O hybridization at the α' -FeRh/MgO interface on the observed PT-TAMR effect.

As presented in Table 1, the PT-TAMR ratio for the α' -FeRh/MgO/Cu junction is calculated to be $\sim 1160\%$ with the same sign as the experimental one but with a much higher value, indicating the great potential of the present PT-TAMR by optimizing α' -FeRh/MgO interface. The experimental PT-TAMR ratio in the present case is only $\sim 20\%$, far below the calculated value, which could be mainly explained by the natural formation of 1 u.c.-thick γ -FeRh at the α' -FeRh/MgO interface. Interestingly, the Rh insertion at the α' -FeRh/MgO interface leads to the reversal of PT-TAMR with the ratio of -73% . The reversal of polarity could be ascribed to the absence of Fe-O hybridization in this scenario (Supplementary Fig. S5).

Table 1 | Transmission and PT-TAMR ratio of α' -FeRh/MgO/Cu and α' -FeRh/Rh/MgO/Cu junctions.

Structure	AFM		FM		PT-TAMR(%) [†]
	T_{maj}^*	T_{min}^*	T_{maj}^*	T_{min}^*	
FeRh/MgO/Cu	2.8×10^{-4}	2.8×10^{-4}	5.4×10^{-6}	3.9×10^{-5}	+1161
FeRh/Rh/MgO/Cu	7.8×10^{-6}	5.0×10^{-5}	1.1×10^{-6}	2.1×10^{-4}	-73

* T_{maj} and T_{min} denote the transmission of majority-spin and minority-spin channels, respectively.

[†] PT-TAMR ratio is calculated by

$$PT - TAMR\% = \frac{(T_{\text{major}} + T_{\text{min}})_{\text{AFM}} - (T_{\text{major}} + T_{\text{min}})_{\text{FM}}}{(T_{\text{major}} + T_{\text{min}})_{\text{FM}}} \times 100\%$$

We add sentences in Page 12 Line 1: **The counterparts for the majority-spin channel are presented in Supplementary Fig. S6** as a link to Note 6, and in Page 9 Line 7: **The DOS of bulk FeRh and interfacial α' -FeRh capped by MgO are calculated in the absence of spin-orbit coupling (Supplementary Fig. S3)** as a link to Note 3.

Accordingly, Notes 6 and 3 are added in the Supplementary material:

Note 6. Transmission distribution in two dimensional Brillouin zone of majority-spin channels

As shown in Fig. S6, the transmission of majority-spin channels for both α' -FeRh/MgO(2.5 u.c.)/Cu and α' -FeRh/Rh(1 u.c.)/MgO(2.5 u.c.)/Cu junctions is lower compared with that of their minority-spin channels (Fig. 4), indicating the negligible role on tunneling conductance. The transmission of both majority-spin and minority-spin channels (Fig. 4a) at AFM state for α' -FeRh/MgO/Cu are identical taking the spin degeneracy into account. Therefore, only the transmission at FM state is shown in Fig. S6a. Given that the induced magnetic moment in interfacial 1 u.c. fcc-Rh breaks the spin degeneracy, besides the FM case, the transmission of majority channels at AFM state for α' -FeRh/Rh/MgO/Cu junctions is also presented in Fig. S6b. Although the transmission at the AFM (Fig. S6b) is stronger than that at FM state (Fig. S6c), both of them are much weaker than their minority counterparts. Thus the tunneling resistance is still determined by the transmission of minority-spin channels (Fig. 4c and d), giving rise to the negative polarity of PT-TAMR in this case.

Figure S6 | Transmission distribution in Two dimensional Brillouin zone for α' -FeRh/MgO(2.5 u.c.)/Cu and α' -FeRh/Rh(1 u.c.)/MgO(2.5 u.c.)/Cu junctions at Fermi level. a, majority-spin channels at FM state for α' -FeRh/MgO/Cu junctions. b,c, majority-spin channels at AFM and FM states for α' -FeRh/Rh/MgO/Cu junctions, respectively.

Note 3. Influence of spin-orbit coupling on DOS of α' -FeRh

Considering that spin-orbit coupling (SOC) is crucial for the previous TAMR effect^{S4}, we also performed DOS calculation of α' -FeRh with SOC at the α' -FeRh/MgO interface. It is found that the DOS variation induced by SOC is negligible when comparing with the DOS difference at AFM and FM states. Thus it is concluded that the origin of the present TAMR is not related to SOC but driven by magnetic phase transition and the resultant change of electronic structure.

Figure S3 | DOS comparison of α' -FeRh at the α' -FeRh/MgO interface with and without spin-orbit coupling. The DOS of α' -FeRh at AFM and FM states with (w) and without (wo) spin-orbit coupling are shown by the solid line and dotted line, respectively.

Response to Reviewer #2

R: Manuscript NCOMMS-17-01166-T entitled "Tunneling magnetoresistance driven by magnetic phase transition " by X. Z. Chen et al. systematically analyzes the PT-TMR effect amplitude as a function of the interfacial microstructure of α' -FeRh/MgO combining fascinating high resolution analytical HRTEM with ab initio DOS calculations. Experimentally this paper is very sound and thorough microstructural, magnetic as well as transport investigations consistently support the main finding. The observed PT-TMR effect amplitude of 20% at room temperature is driven by the magnetic phase transition of α' -FeRh in addition to the presence of a naturally formed one-unit cell thick γ -FeRh layer at the α' -FeRh/MgO interface, which results in an opposite polarity as expected for α' -FeRh. Furthermore, the authors present two approaches – annealing and inserting an additional γ -FeRh layer at the α' -FeRh/MgO interface – successfully avoiding the oxidation problematic. Both approaches repair the reversed polarity of the PT-TMR effect amplitude but result in a PT-TMR effect amplitude reduced to about -5% at maximum only. All discussions presented in this manuscript are taking into account all aspects of current and previous literature.

In summary, this is a very decent work but dealing with the common problematic known for all TMR-cells, the affinity for oxygen of elements of the magnetic electrodes sandwiching an oxidic barrier. Since both approaches to avoid the oxygen diffusion are drastically reducing the initial TMR-effect amplitude, I cannot see at all evidence that this finding is adding a different dimension to MRAM and spintronics.

A: Thanks for the referee's positive evaluation on the scientific side. We admit that the demonstrated TAMR at room temperature is only ~20% driven by the magnetic phase transition in the present devices. But this value is a promising progress even from the applicative viewpoint because the previous TAMR is lower than 1% at room temperature or even generally limited at low temperature (<100 K). More interestingly, the calculation on the transmission of the α' -FeRh-based junctions shows a large transmission difference for the AFM and FM state, giving rise to the PT-TAMR ratio exceeding 1000% with perfect α' -FeRh/MgO interface (Fig. 4, Table 1). Thus it is promising to achieve high PT-TAMR ratio for spintronics application, as predicted theoretically, by eliminating the formation of γ -FeRh at the α' -FeRh/MgO interface. This might be fulfilled by improved growth of α' -FeRh and MgO with optimized sputtering or molecular beam epitaxy. The great potential of the enhancement of the PT-TAMR ratio would add a different dimension to MRAM and spintronics. Meanwhile, the absolute value of PT-TAMR ratio reduces seriously to ~70%

when one unit cell-thick of fcc-Rh is inserted at the α' -FeRh/MgO interface (Fig. 4, Table 1), indicating the profound influence of the unexpected phase at the interface on the tunneling effect and making the comparable ratio of 20% in our devices understandable.

Both experimental approaches of annealing and inserting an additional γ -FeRh layer not only increase the distance between α' -FeRh functional layer and tunneling interface barrier but also block the Fe-O hybridization (Fig. S5), leading to the reduced PT-TAMR ratio and the reversal of the PT-TAMR polarity. Thus, the rich interfacial manipulation reveals that the present PT-TAMR is controllable and designable, guaranteeing the fundamental significance of our work.

We revised the sentences in the discussion part in Page 16: **According to the remarkable transmission difference between the AFM and FM states (Table 1), a larger PT-TAMR ratio of hundreds percent is highly warranted in α' -FeRh-based MTJs if higher quality α' -FeRh/MgO interface was obtained. This might be achieved by optimizing growth parameters or by other growth techniques, e.g., molecular beam epitaxy, to satisfy the requirements of magnetic random access memory on the PT-TAMR ratio. Meanwhile, the memory driven by magnetic phase transition has the potential to be operated in ultrafast dynamics, because the structural evolution of FeRh is faster than the magnetic response²⁹.** For more details on the calculations of transmission distribution and corresponding PT-TAMR ratio, please refer to our answer to the second question of Reviewer #1.

Response to Reviewer #3

R: The authors report interesting experimental results of demonstration of tunneling magnetoresistance (TMR) using the first order magnetic phase transition between antiferromagnetism and ferromagnetism in α' -FeRh at room temperature. They have proved that the electronic structure near the tunneling barrier has a crucial role for determining the TMR ratio. If we can properly control the phase transition of this material near the tunneling interface, this method will be very useful. In my opinion, their idea is promising for the room temperature applications. However, the following issues should be properly addressed in the manuscript for publication in Nature Communications.

A: We are grateful to Reviewer #3 for the positive evaluation.

R: 1. They did not mention whether this device can have a non-volatile memory function. In Fig. 2c, the resistance area (RA) always returns to the low state when the magnetic field is switched off. For the memory applications, we have to maintain each state without a magnetic field.

A: According to the referee's suggestion, we show thermal assisted nonvolatile function in the absence of magnetic field in Fig. S8.

We add a sentence in Page 16 Line 12 for Note 8: **it is worthy pointing out that temperature variation (Supplementary Fig. S8) or large magnetic field (several Tesla) is not indispensable for the PT-TAMR.**

Note 8 is added in the Supplementary material:

Note 8. Thermal assisted nonvolatile function

Given that the magnetic phase transition of FeRh is the first order phase transition with a hysteresis window, the PT-TAMR effect naturally shows a nonvolatile memory function. Figure S8 displays that two different resistance states of α' -FeRh/MgO/ γ -FeRh junctions could be preserved at zero-field after superheating and undercooling processes. For this measurement, superheating was carried out with the temperature increasing to 400 K followed by cooling back to 330 K, a temperature almost locating at the middle of the hysteresis window for the AFM-FM transition at zero-field. In this case, α' -FeRh keeps at FM state and a higher RA for 30 min (the black line in Fig. S8). Similarly, when the junction was cooled down to 250 K and then warmed back to 330 K, it stays at AFM state accompanied by a stable low resistance for 30 min (the red line in Fig. S8). Three circles of such measurements were carried out to confirm that the nonvolatile function of the PT-TAMR effect is reproducible.

Figure S8 | Resistance area (RA) recorded at 330 K and zero-field after superheating and undercooling. α' -FeRh at FM and AFM states after

superheating and undercooling correspond to high (black line) and low resistances (red lines) of α' -FeRh/MgO/ γ -FeRh junctions, respectively. Each line is composed by 300 counts recorded in 30 min. Three circles of such measurements are shown in the figure.

R: 2. They did not mention the effect of the strain induced by the phase transition. It may degrade the endurance of the device.

A: The volume of α' -FeRh is increased by $\sim 1\%$ by the magnetic phase transition from AFM to FM state. Nevertheless the lattice is almost expanded in the *c*-axis direction, while the in-plane lattice of α' -FeRh keeps almost constancy due to the clamping effect from MgO substrate (Phys. Rev. Lett. 109, 117201 (2012)). Therefore, the structure of tunneling interface (α' -FeRh/MgO) should suffer little from the lattice expansion induced by the magnetic phase transition. In Fig. S2 we show magnetic field dependent resistance ($RA-\mu_0H$) in ten circles, which keeps almost unchanged, indicating the good endurance of the present devices.

We add a sentence in Page 7 Line 5 from the bottom for Note 2: **the PT-TAMR ratio keeps almost unchanged, revealing that the present PT-TAMR effect is stable, repeatable, and reproducible (Supplementary Fig. S2).**

Note 2 is added in the Supplementary material:

Note 2. Endurance of the PT-TAMR effect

The strain effect induced by the magnetic phase transition might result in the degradation of the PT-TAMR effect. Nevertheless the lattice is almost expanded in the *c*-axis direction, while the in-plane lattice of α' -FeRh keeps almost constancy due to the clamping effect from MgO substrate^{S3}. Therefore, the structure of tunneling interface (α' -FeRh/MgO) should suffer little from the lattice expansion induced by the magnetic phase transition. Figure S2 shows the magnetic field dependent resistance ($RA-\mu_0H$) at 300 K in ten circles, which keeps almost unchanged, indicating good endurance of the present devices in ten circles.

Figure S2 | Magnetic field dependent resistance ($RA-\mu_0H$) at 300 K in ten circles.

R: 3. I hope that the authors describe the switching speed. I wonder if it is slower than the magnetization switching.

A: The doubt by the Referee is reasonable. It is easy to think that the switching speed of magnetic phase transition is slower than the magnetization switching. However, on the basis of the laser-induced magnetic phase transition of FeRh measured by time-resolved x-ray diffraction and magneto-optical Kerr effect, it is demonstrated that structural response of ferromagnetic domain nucleation (~ 30 ps) is faster than magnetic response of moment realignment (~ 60 ps) (Phys. Rev. Lett. 108, 087201(2012)). Therefore, the memory relies on the magnetic phase transition have the potential to be operated in ultrafast dynamics. Accordingly, we add a sentence in Page 16 Line 10: **Meanwhile, the memory driven by magnetic phase transition has the potential to be operated in ultrafast dynamics, because the structural evolution of FeRh is faster than the magnetic response²⁹.**

R: 4. The physical mechanism of the change of the density of states (or the electronic structure) by the magnetic phase transition should be explained in more detail. This is important to increase the TMR ratio.

A: Thanks for the referee's helpful suggestion. For bulk α' -FeRh, each magnetic state (AFM and FM) has a fixed electronic structure, as characterized by hard x-ray photoemission (Phys. Rev. Lett. 108, 257208 (2012)). Differently, the electronic structure of the interfacial α' -FeRh at each magnetic state could be modulated with the interfacial effect, such as Fe-O hybridization at the α' -FeRh/MgO interface. Besides the total DOS of one Fe atom and Rh atom at the α' -FeRh/MgO interface (Fig. 3c), electronic structure calculations of the Fe and O atom in the vicinity of the α' -FeRh/MgO interface in Fig. S5 present that the DOS is greatly enhanced at the Fermi level for AFM

α' -FeRh, leading to the lower resistance of the AFM state compared to its FM counterpart. This finding suggests that the Fe-O hybridization at the interface plays a profound effect on the positive polarity of the PT-TAMR, in contrast to the case of bulk α' -FeRh.

To quantitatively understand the present PT-TAMR effect, the calculations of transmission distribution and PT-TAMR ratio were carried out. Interestingly, the positive (~1160%) and negative (-73%) PT-TAMR ratios are obtained for α' -FeRh/MgO/counter-electrode and α' -FeRh/Rh/MgO/counter-electrode junctions (Fig. 4 and Table 1), respectively. Their polarity coincide with the experimental observation, 20% and -3% PT-TAMR ratios for the as-grown and interfacial engineered (intended γ -FeRh insertion or annealing) junctions, but with a much higher magnitude, most likely due to the “perfect” α' -FeRh/MgO interface without any γ -FeRh for the calculation. This comparison indicates that the present α' -FeRh-based junctions are promising with optimized α' -FeRh/MgO interface for application in turn. For more details on the calculations of transmission distribution and corresponding PT-TAMR ratio, please refer to our answer to the second question of Reviewer #1.

We add a sentence in Page 10 Line 4 from the bottom: **whose DOS is inverted by contacting oxides (Fig. 3c) and resultant Fe-O hybridization (Supplementary Fig. S5).**

Note 5 is added in the Supplementary material:

Note 5. Enhancement of DOS at AFM state due to Fe-O hybridization

Figure S5 illustrates the α' -FeRh/MgO supercell and corresponding DOS for the Fe and O atoms in the vicinity of the α' -FeRh/MgO interface when α' -FeRh is at AFM state. The most eminent feature in the figure is the enhanced DOS at the Fermi level for both Fe1 and O1 (Fig. S5b and S5c), which locates at the first neighbor of the α' -FeRh/MgO interface, as marked in Fig. S5a. However, such a feature vanishes for Fe2 and O2, which is one unit cell away from Fe1 and O1, respectively. This comparison demonstrates that Fe-O hybridization plays a profound role on the enhancement of DOS of Fe for AFM α' -FeRh. Such a high DOS for AFM α' -FeRh leads to the low tunneling resistance of AFM α' -FeRh and the resultant positive PT-TAMR of α' -FeRh-based junctions, as shown in Fig. 3. On the other hand, the PT-TAMR effect is so sensitive to the α' -FeRh/MgO interface that both the polarity and amplitude of the PT-TAMR could be strongly manipulated by the design of α' -FeRh/MgO interface.

Figure S5 | DOS for the Fe and O atoms in the vicinity of the α' -FeRh/MgO interface when α' -FeRh is at AFM state. a, α' -FeRh/MgO supercell used for DOS calculation. b, Comparison of DOS for O1 and O2 atoms, in which O1 is at the first neighbor of the α' -FeRh/MgO interface and O2 is one unit cell away from O1. c, Comparison of DOS for Fe1 and Fe2 atoms, in which Fe1 is at the first neighbor of the α' -FeRh/MgO interface and Fe2 is one unit cell away from Fe1.

R: 5. Why is the tunnel resistance background changed when the temperature is changed? If the tunneling effect is dominant, it should be constant.

A: In general, for the tunneling magnetoresistance, the resistance in parallel state is nearly constant while the resistance in antiparallel state decreases with increasing temperature (Nature Mater. 3, 862 (2004)). It is proposed that as temperature increases, the enhanced magnetic disorder would increase and decrease resistance in parallel and antiparallel states, respectively. On the other hand, thermal excitations across the barrier would decrease the resistance in both states when heating the junctions (Nature Mater. 3, 862 (2004)). These two factors together would lead to constant background of tunneling resistance. However, in TAMR device with only one magnetic electrode, thermal excitations across the barrier could be the main contribution to the resistance background, leading to its drop when enhancing temperature. This is corroborated by the reduced resistance background of IrMn/MgO/Ta junctions with increasing temperature (Appl. Phys. Lett. 102, 192404 (2013), Nature Mater. 10, 347 (2011)). We add a sentence in Page 6 Line 5: **Note that**

the resistance background of PT-TAMR decreases with increasing temperature, which might be due to the thermal excitations across the barrier^{5,22}.

New References in the revised version

22. Petti, D. *et al.* Storing magnetic information in IrMn/MgO/Ta tunnel junctions via field-cooling. *Appl. Phys. Lett.* 102, 192404 (2013).

27. Chantis, A. N., Belashchenko, K. D., Tsymbal, E. Y. & van Schilfgaarde, M. Tunneling anisotropic magnetoresistance driven by resonant surface states: first principles calculations on an Fe (001) surface. *Phys. Rev. Lett.* 98, 046601 (2007).

28. Chantis, A. N. *et al.* Reversal of spin polarization in Fe/GaAs (001) driven by resonant surface states: first-principles calculations. *Phys. Rev. Lett.* 99, 196603 (2007).

29. Mariager, S. O. *et al.* Structural and Magnetic Dynamics of a laser induced phase transition in FeRh. *Phys. Rev. Lett.* 108, 087201 (2012).

34. Giannozzi, P. *et al.* QUANTUM ESPRESSO: a modular and open-source software project for quantum simulations of materials. *J. Phys.: Condens. Matter* 21, 395502 (2009).

S3. Bordel, C. *et al.* Fe spin reorientation across the metamagnetic transition in strained FeRh thin films. *Phys. Rev. Lett.* 109, 117201 (2012).

S4. Park, B. G. *et al.* Tunneling anisotropic magnetoresistance in multilayer (Co/Pt)/AlO_x/Pt structures. *Phys. Rev. Lett.* 100, 087204 (2008).

REVIEWERS' COMMENTS:

Reviewer #1 (Remarks to the Author):

The authors have provided a comprehensive response to questions of all three referees, including additional calculations and much improved interpretation of their experiment based on them. I think that 1000% (and much less observed, but it could be optimized like it was the case of conventional TMR in Fe/MgO/Fe junctions) finding without spin-orbit coupling playing a crucial role is interesting enough to justify publication in Nature Communications. For example, very recent analysis <https://arxiv.org/abs/1701.00462> of strongest possible spin-orbit coupling introduced by 3D topological insulator (TI) in normal-metal/TI/ferromagnetic-metal junctions find maximum TAMR of only 60%.

Reviewer #3 (Remarks to the Author):

I think that the authors have well responded to the reviewers' comments. Their experimental results are very interesting, so I recommend publication of their manuscript.

Response Letter of NCOMMS-17-01166A

Response to Reviewer #1

R: The authors have provided a comprehensive response to questions of all three referees, including additional calculations and much improved interpretation of their experiment based on them. I think that 1000% (and much less observed, but it could be optimized like it was the case of conventional TMR in Fe/MgO/Fe junctions) finding without spin-orbit coupling playing a crucial role is interesting enough to justify publication in Nature Communications. For example, very recent analysis <https://arxiv.org/abs/1701.00462> of strongest possible spin-orbit coupling introduced by 3D topological insulator (TI) in normal-metal/TI/ferromagnetic-metal junctions find maximum TAMR of only 60%.

A: We appreciate the positive evaluation of Reviewer #1.

Response to Reviewer #3

R: I think that the authors have well responded to the reviewers' comments. Their experimental results are very interesting, so I recommend publication of their manuscript.

A: Thanks for the positive evaluation of Reviewer #3.